# CRISPR-Cas9 System: A Prospective Pathway toward Combatting Antibiotic Resistance

**DOI:** 10.3390/antibiotics12061075

**Published:** 2023-06-19

**Authors:** Muhammad Uzair Javed, Muhammad Tahir Hayat, Hamid Mukhtar, Kalman Imre

**Affiliations:** 1Department of Biotechnology, Quaid-i-Azam University, Islamabad 45320, Pakistan; 2Institute of Industrial Biotechnology, Government College University, Lahore 54000, Pakistan; 3Department of Animal Production and Veterinary Public Health, Faculty of Veterinary Medicine, University of Life Sciences “King Mihai I” from Timişoara, 300645 Timișoara, Romania

**Keywords:** antibiotics resistance genes, target sequence, genome editing, guide RNA, drug resistance

## Abstract

Antibiotic resistance is rising to dangerously high levels throughout the world. To cope with this problem, scientists are working on CRISPR-based research so that antibiotic-resistant bacteria can be killed and attacked almost as quickly as antibiotic-sensitive bacteria. Nuclease activity is found in Cas9, which can be programmed with a specific target sequence. This mechanism will only attack pathogens in the microbiota while preserving commensal bacteria. This article portrays the delivery methods used in the CRISPR-Cas system, which are both viral and non-viral, along with its implications and challenges, such as microbial dysbiosis, off-target effects, and failure to counteract intracellular infections. CRISPR-based systems have a lot of applications, such as correcting mutations, developing diagnostics for infectious diseases, improving crops productions, improving breeding techniques, etc. In the future, CRISPR-based systems will revolutionize the world by curing diseases, improving agriculture, and repairing genetic disorders. Though all the drawbacks of the technology, CRISPR carries great potential; thus, the modification and consideration of some aspects could result in a mind-blowing technique to attain all the applications listed and present a game-changing potential.

## 1. Introduction

Antibiotics have had a remarkable effect on human and animal health since their discovery. Yet, the widespread use of disinfectants and antibiotics has resulted in unparalleled health issues around the world. This condition started the moment microbes started developing resistance mechanisms against lingering antibiotics [1]. The word ‘antibiotic-resistance’ is the capability of bacteria to avoid being treated or blocked via antibiotics. There has been a constant concern about bacterial resistance to antibiotics since the beginning of antibiotic growth. Bacterial resistance in the case of pathogenic bacteria leads to one of the most serious bacteria-caused risks because it induces fatal infections and causes prolonged disease, high financial costs, and increased morbidity [2]. Despite the fact that antimicrobials have had a significant effect on Western medicine since their discovery in the early nineteenth century, resistant bacteria have evolved. Some of the clinical bacteria have been shown to be resistant to all available antibiotics, jeopardizing all advances achieved during the antibiotic era and necessitating the introduction of new therapies [3].

Antimicrobial resistance genes (ARGs), which are naturally occurring genetic elements, play a crucial role in the development and spread of antibiotic resistance. However, mistreating antibiotics in humans and animals is also elevating the process. Exposure to antibiotic molecules in clinical and agricultural settings and many countries’ involvement in water reuse have escalated the antibiotic resistance problem, particularly in the Middle East and North Africa [4]. Surprisingly, a limited number of bacteria, known as ESKAPE pathogens, are accountable for common antibiotic-resistant diseases. According to the definition by the Infectious Diseases Society of America, ESKAPE pathogens are a class of antibiotic-resistance bacteria that are in turn accountable for making the cure of nosocomial infections problematic. These pathogens include *Pseudomonas aeruginosa*, *Enterobacter* spp., *Staphylococcus aureus*, *Acinetobacter baumannii*, *Klebsiella pneumoniae*, and *Enterococcus faecium* [5].

The inherent challenges in discovering new antibiotics, along with a lack of financial incentives, have ensured that the discovery of novel antibiotics is a slow process. Furthermore, the stride of development of novel antibiotics lacks behind the stride of evolving antibiotic resistance, which is inadequate to tackle the upsurge in antibiotic resistance. As an outcome, genomic engineering approaches for gene knock-in and knock-out of sequence-specific DNA antibiotic targets have been projected as new boulevards for reducing and restraining antibiotic resistance in pathogens. These techniques include zinc finger nuclease (ZFNs), phage therapy application of peptide nucleic acid (PNA) as an ultra-fine-range antibiotic and collected frequently interspaced short palindromic repeat- CRISPR-associated (CRISPR Cas) systems [6].

Other techniques, such as PNA antibiotics, ZFNs, and phage therapy, show potential in combating antibiotic resistance, though they still have some significant limitations. PNA antibiotics, for instance, have restricted target specificity and can potentially damage healthy microbiota [7]. Phage therapy, on the other hand, necessitates the discovery and use of certain bacteriophages to target resistant bacteria, which makes it a labor-intensive and complicated operation [8]. ZFNs also have restricted target specificity and can cause off-target effects, leading to unintended genetic changes [9]. On the contrary, CRISPR-Cas9 has proven to be a potential technique in case of antibiotic resistance because of its efficient genome editing capabilities and highly specific targeting. It permits precise bacterial gene modifications, including those responsible for antibiotic resistance. CRISPR-Cas9 is also very adaptable and can be used to target a variety of bacterial strains, making it a highly versatile tool in combating antibiotic resistance [10].

## 2. Overview of CRISPR Cas9

The CRISPR system was initially identified in the DNA sequences of *Escherichia coli* bacteria and was described by Ishino et al. [11] of Osaka University (Japan) in 1987. However, it was not until 2007 that the potential of the CRISPR system for gene editing was realized when the first experimental information about the mechanism of action of the CRISPR system was obtained by two French food scientists named Rodolphe Barrangou and Philippe Horvath working with yogurt cultures of bacteria *Streptococcus thermophilus* for the Danish company Danisco [12]. This system functions by utilizing RNA molecules to direct the protein Cas9 to a specific site in a genome, where it can then cleave the DNA at that location, thereby enabling researchers to precisely add, delete, or modify specific genes [13].

The CRISPR system is primarily acquired through horizontal gene transfer including transformation, conjugation, and transduction. Bacteria and archaea can acquire CRISPR sequences from other bacteria and archaea via this process [14]. Upon acquisition, these sequences become integrated into the genome of the bacterium or archaeon and can be utilized to defend against foreign genetic material, such as viruses. CRISPR-Cas represents an adaptive immune system that is present in the majority of archaeal and bacterial species. This system confers protection against infections caused by viruses, phages, and other foreign genetic elements [15]. Notably, this fascinating system is present in nearly 87% of archaeal genomes and 50% of bacterial genomes [16].

The CRISPR-Cas operons are genetic elements that occur in bacteria and archaea and serve as an adaptive immune system. This system comprises a collection of short, repeated DNA sequences (known as CRISPRs) separated by spacer regions that correspond to viral or plasmid DNA sequences. Upon subsequent infections, the CRISPR-Cas system can identify and break down these exogenous genetic elements [17].

This genome editing system consists of a guide RNA molecule that directs the nuclease to a particular genomic target site of the genome and a Cas9 nuclease [18]. To make the mature gRNA, the endogenous bacterial machinery processes a single chimeric guide RNA (sgRNA) that is comprised of a combination of a CRISPR RNA (crRNA) and a fixed trans-activating crRNA (tracrRNA) [19]. The genetic loci of CRISPR-Cas systems have the CRISPR array that consists of similarly flanking sequences (spacers) and short repeated sequences (repeats). Protospacers are used as spacers in CRISPR arrays. These protospacers are in turn derived from DNA sequences from infecting plasmid or phage. The most important functional elements in CRISPR systems are Cas proteins encoded upstream of the CRISPR array and control system operation [20,21]. 

Class 1 and Class 2 are the two main classes in which the CRISPR-Cas system is categorized. There are six types (I to VI), and multiple subtypes of the CRISPR-Cas system, with Class 1 systems (Type I, III, and IV) having multi-Cas protein effector complexes and Class 2 systems (Type II, V, and VI) having a single effector protein [22,23]. Table 1 summarizes the classification, standard features, and representative members of each CRISPR-Cas system.

## 3. The Perilous Persistence of Drug Resistance

Microorganisms are extremely intelligent and have evolved strategies to proliferate quickly and survive under unfavorable conditions. Antibiotic resistance emerged shortly after antibiotics were first used in clinical practice, but the phenomenon was slow to emerge and was largely dismissed as a minor concern [29]. *Streptococcus pyogenes* resistance to Sulfonamide emerged in human medical situations at the start of the 1930s, whereas *S. pyogenes* resistant to penicillin was observed in the 1940s. In the 1950s, the appearance of multidrug-resistant bacteria was illustrated [1]. Antibiotic tolerance is thought to evolve in two ways: vertical evolution and horizontal evolution. Horizontal evolution refers to the acquisition of antibiotic-resistance genes from other bacteria by conjugation, transduction, or transition, while vertical evolution refers to mutations responsible for antibiotic tolerance, which in turn can be passed on to offspring. Horizontal gene transfer is an essential method for transmitting antibiotic-resistant genes amongst microorganisms, as per a detailed genomic study of human and animal pathogens [30].

Antibiotic resistance was not initially recognized as a serious problem, despite the potential consequences it could have on public health. However, the discovery of bacteria bearing broad spectrum b-lactamases (ESBLs) conferring resistance to cephalosporins and penicillins, multidrug-resistant *Neisseria gonorrhoeae*, *Pseudomonas aeruginosa*, *Acinetobacter baumannii*, and *Enterobacteriaceae*, and extensively drug-resistant (XDR) *Mycobacterium tuberculosis* drew the attention of clinicians and biochemists [31,32]. The New Delhi metallo-b-lactamase 1 (NDM-1) was initially discovered in a *Klebsiella pneumoniae* and *Escherichia coli* isolate of a patient in a New Delhi, India, hospital a decade ago [33]. 3rd generation cephalosporin-resistant *E. coli* infections are the most common, accounting for more than half of all infections in the population. This means that antimicrobial stewardship should not be constrained to a hospital to lessen the trouble of antimicrobial resistance but must include prescribers and primary care interventions.

Antibiotic resistance may be on the rise as a result of failing public health services, the proliferation of irresistible infections, and the misuse of antibiotics without proper medical guidance [34]. In developed or developing nations, where medical facilities are scarce, the threat is amplified. Furthermore, unrestricted access to antibiotics in pharmacies, coupled with a lack of consumer knowledge of the consequences of misuse, has led to a prevalence of self-medication practices [35]. 

## 4. Roles of CRISPR-Cas in Antibiotic-Resistant Bacteria

As a bacterial adaptive immune system, the clustered regularly interspaced short palindromic repeats–CRISPR-associated (CRISPR-Cas) system is recognized as a developing way to regulate antibiotic-resistant. When applied against bacterial genomic sequences, this system’s programmable Cas nuclease could be lethal or aid in diminishing antibiotic resistance in bacteria. CRISPR-Cas methods could pave the way for developing innovative antibiotics able to get rid of multidrug-resistant (MDR) infections and distinguish them between valuable and pathogenic bacteria. These methods can be utilized to statistically and specifically eradicate particular bacterial species based on their sequencing, opening up opportunities in the therapies of MDR infection, microbial consortia research, and industrial fermentation control [6].

In several research studies, the CRISPR-Cas mechanism showed a clear negative association with antibiotic resistance in certain bacteria, for example, *Enterococci* [36], but not in others, such as *E. coli*. The three CRISPR loci present in *Enterococci* are CRISPR1-Cas, orphan CRISPR2, the non-existence of Cas genes, and CRISPR3-Cas. According to the outcomes, orphan CRISPR2 is present in every *E. faecalis* strain, while expression of CRISPR1-Cas as well as CRISPR3-Cas differs between strains. Furthermore, a discovery was carried out which depicted that CRISPR1-Cas and CRISPR2 are functionally connected. Table 2 depicts the therapeutic applications of several CRISPR-Cas systems.

According to CRISPR spacer analysis, pheromone-response plasmids have a key function in the plasticity of the *Enterococcal* genome, mobilizing the virulence proteins that are chromosomally encoded, antibiotic-resistant, and can boost their transfer and displace CRISPR-Cas [45]. The transfer of the pheromone-responsive plasmid spacer revealed the fact that specific elements have a proclivity for being merged into CRISPR loci as *E. faecalis* regularly encounters spacers, but no CRISPR spacers against Tn916 have yet been found, which might be used to transmit antibiotic resistance in *Enterococci* [46,47].

The tetM gene, which is usually distributed through Tn916 and some other conjugative transposons, is found in *E. faecalis* with CRISPR loci, suggesting that conjugative transposons can get through this defense. Furthermore, the Inc18 plasmid family, which has transmitted the vancomycin-resistance gene by *Enterococci* to MRSA, lacks a spacer. These findings could be due to the inefficiency of element transfer, which lacks a mechanism for successful frequency of inter-species transition, or pair forming. Furthermore, similar functions were determined in the CRISPR-positive *Streptococcus thermophiles* strain that gained additional spacers generated by the virus, which made it immune to the infection by phage [12].

Brouns et al. discovered that Lambda phage sensitivities are lower in the *E. coli* K12 strain with an engineered CRISPR-Cas technique with a spacer that affects Lambda phage genes [48]. In addition, Maraffini and Sontheimer showed that the CRISPR-Cas system would prevent plasmid conjugation in *S. epidermidis*, suggesting that the CRISPR-Cas system has a wider and an essential function during inhibition of horizontal gene transfer (HGT) [49]. CRISPR1, CRISPR2, CRISPR3, and CRISPR4 are the four CRISPR loci present in *E. coli*, each with a different form of the Cas gene [50,51]. Since all *Escherichia* strains that bear CRISPR1 lack CRISPR4, no *Escherichia* strain genome has more than three CRISPR. Touchon et al. [50] found that the existence of plasmids, integrons, or antibiotic resistance has little effect on the number of repeats in *E. coli* CRISPR loci. They also found no correlation between the presence of spacers matching antibiotic resistance genes or elements associated with the movement of antibiotic resistance genes, such as Tn3, intI, ISEcp1, ESBL production, ESBL type, or replicon. Additionally, no spacers matching plasmid sequences were detected. [50]. Only a few spacers matching plasmid genes were found in the strains of *E. coli* that were analyzed, and the spacers were unique to every CRISPR type, with one strain possessing 15 spacers; 77% of the strains had one, 13.2% had two, and 7.5% had three or four spacers. Touchon et al. also discovered that CRISPR has no impact on the plasmid epidemiology in *E. coli* or the transfer of antibiotic-resistant genes. In *Enterococci*, CRISPRs are linked to antibiotic resistance in reverse. Another study describes the role of CRISPR-positive plasmids in the spread of antibiotic-resistance genes (ARGs) and virulence genes (VGs) in the *Klebsiella* genus. Plasmids that are CRISPR-positive have been found to engage in intense competition with other plasmid types, particularly conjugative plasmids. This competition has been shown to have an impact on the way in which plasmids are transmitted [52].

According to the findings, the CRISPR-Cas system could have a diverse impact on antibiotic resistance in various species. This is due to differences in the CRISPR-Cas system’s evolutionary histories, CRISPR locus formation by incomplete or complete deletion of the Cas genes cluster, and the existence of anti-CRISPR proteins, similar to Toxin-Antitoxin, or restriction and modification practices [53]. When the Cas system is completely developed, the number of repeats is high-level. Similarly, when the Cas system is gradually eroding, the number of repeats is moderate; when only Cas system remnants are visible, the number of repeats is minimal, when the Cas system is fast eroding [54]. Interestingly, a large number of repeats in three closely related *E. coli* genomes, namely BL21, BL21-DE3, and B-REL606, are absent of Cas genes, while *E. coli* strains SMS35 have no CRISPR2 owing to a recent sucrose operon insertion [54].

The number of repeats is a helpful marker for a system’s future efficiency and credibility. Furthermore, if a mobile part is acquired and the CRISPR locus has spacers that complement the chromosome, the host could be put in grave danger. Chromosome degradation could occur when the CRISPR-Cas system induces DNA plasmid breakdown, which involves the spacer identical to chromosomal DNA. Furthermore, even though the CRISPR-Cas mechanism merely interferes with gene expression, the foreign factor may still manipulate the host. Bacterial genomes may have formed a defense mechanism to combat the attacking CRISPR-Cas system as a result. This method could include the application of native CRISPR. This strategy could include using native CRISPR as an anti-CRISPR, as shown in *E. coli* [52] and *P. aeruginosa* [53,54].

## 5. Neutralization of Antibiotic-Resistant Genes by CRISPR-Cas System

The direct targeting and cleaving of DNA are carried out by the programming of Cas protein which allows the flanking of RNA-based spacers through the partial repeats in order to encode the conforming protospacers. Consequently, upon the fundamentals which are present in the CRISPR array, the respective system is instructed for the targeting and cleaving of any sort of DNA under in vivo conditions. This array has been formerly utilized in the targeting of the respective bacteria which carry exclusive genes pertaining to encoding antibiotic resistance [55,56]. The latest research has concluded that the targeting which is carried through the CRISPR-Cas system has resulted in being cytotoxic. This cytotoxicity results in cell death, which is provoked by the irreversible chromosomal defects which are introduced as a result of the system [57,58]. Figure 1 illustrates the action mechanism of CRISPR-Cas system.

Due to the limitation of the CRISPR-Cas system to the domains of archaea and bacteria, the separation, optimization, and production of the CRISPR-Cas transfer carriers and vectors is quite significant for the implementation of RNA-guided nucleases which carry the ability to target new strains for instance, multi-drug resistant (MDR) pathogens and other crucial elements of the endogenous microbiota [57]. Moreover, the regulation of specific genes in wild-type populations (for instance, virulence factors and antibiotic resistance genes) by the RNA-guided nucleases might occur due to the delivery paths which are utilized in the higher and more complex species [59]. The delivery of the Cas 9 nuclease to the microbial communities, which ought to target the exclusive DNA sequences of antibiotic resistance and bacterial pathogens, is carried by polymer-derivatized CRISPR-nanocomplexes36, bacteria with conjugable plasmids [38,57], and/or bacteriophages [39,57]. The CRISPR-Cas system of *E. coli* is typed I-E. It encodes about six genes in two operons, comprising Cas ABCDE and Cas 3, to particular and significant sites within the genomes of variant sequences [48].

Consequently, it was summarized that the dominant removal can be attained by targeting various and diverse loci at the same time. The diverse array which is to be targeted is inclusive of msbA, asd, ftsA, and nusB, with the addition of targeting multiple sites at the same time, which are near to those depicted by the extent of elimination, can be acquired [60]. Instead of using an influential system for the deliverance of type I, CRISPR-Cas system inside the bacterium, they utilized a metamorphosis in the stated work inclusive of targeting the chromosomal genes exclusive to a subset of the mixed population and resistance genes present on the extrachromosomal parts. The utilization of Cas-9 phagemid was carried out for the elimination of MRSA strains from the mixed population of bacteria, which pertains to a plasmid envisioned to be encapsulated in phage capsids [61]. Target-specified CRISPR-Cas9 plasmids which are intended to point out tetracycline-resistant plasmids such as pUSA01 and pUSA02, were also synthesized. To carry out the treatment against the clinical isolate of *S. aureus* USA300Φ, a phagemid was synthesized using Cas9 and crRNA obtained from the methicillin resistance gene mecA90 (pDB121: mecA).

Cell death remained unobserved in any of the Cas9 plasmids that target tetracycline-resistant plasmid, but tetracycline sensitivity, on the other hand, was observed in more than 99.99% of the cells. The fraction of *S. aureus* USA300 has reduced from 50% prior to treatment to 0.4% following treatment with pDB121: mecA phagemid. They also built enterotoxins eugene91in the phagemid and found that it was capable of killing all strains of *S. aureus* with comparable efficiency. Citorik et al., employed two techniques to introduce the Cas9 nucleases into bacteria, including conjugative plasmid and M13-based phagemid, to target blaNDM-1and blaSHV-18, that encode pan-resistant to beta-lactams and extended-spectrum antibiotic resistance, correspondingly [62]. The phages were used to treat reactive strains and viable cell counts were reduced by 2–3 logs.

Another demonstration was carried out, which stated that the CRISPR-Cas apparatus is capable of variating among the tolerant and the susceptible strains. Moreover, the programmed phagemid gyrA pertained to be cytotoxic to the following stated specific mutation; quinolone resistance-causing chromosomal gyrA mutations, which states that the wild-type gyrA gene present in isogenic and *E. coli* are not cytotoxic [57]. Another non-viral delivery approach was developed with its fundamentals relying upon Cas9-nanocomplex, including Cas9, sgRNA targeting mecA, and branched polyethyleneimine, a cationic polymer (bPEI). The implementation of the bPEI was subjected to the packaging of gRNA which results in an increment in the deliverance of Cas9 to the MRSA strains. This carrier is the most common which is usually implemented for the delivery of the genes [63,64]. The conjugation of Cas9 which is carried out with the bPEI has an increased rate of absorption in the bacterium as compared to the native Cas9 which is joint with bPEI and also Cas9 which is blended with lipofectamine which is a carrier for the transport of genes in the mammalian cells. The latter subjects did not depict any increment in the activity as compared to the former one [65,66].

A hypothesis was formulated that the respective occurrence is because of the increment in the protein polarity or by the high cationic properties of the polymer of bPEI [67]. Moving on, the treatment of cultured MRSA strains with Cas9-bPEI depicts inhibition of growth in 6 g/mL oxacillin-containing agar media. However, the untreated germs did not show any such inhibition. Without the inclusion of sgRNA in the treatment through the Cas9-bPEI complex resulted in a reduction of 32% of growth in comparison to the inclusion of sgRNA in the Cas9-bPEI complex. The latter group was added as a control in order to come around the difference created in the two scenarios. However, the observation of native Cas9/sgRNA counteracted with a lipofectamine carrier caused no reduction in the growth [67]. The breakthrough stated above might be considered a step ahead in the development of a CRISPR-based antimicrobial drug and subsequently a vector-free CRISPR transmission. The introduction of such a technology will lead to the prevention of off-target effects and immunogenicity difficulties. Moreover, these might prove to be a phenotypic improvement and the modification of the bacterial genome.

The reduction in the frequency of conjugation which is attainable by the bacteriophage- or phagemid-mediated transport occurs as a limiting factor in the CRISPR-mediated killing through plasmids. Despite this, the conjugative plasmid transmission of CRISPR nucleases is rendered as a potent option as no cellular receptor is required for conjugative plasmids [68]. The building of the straightforward huge coding capacities depicts resistance against the restriction modification system acquiring a wide range of hosts [69]. The transfer rate of the conjugative plasmid which encodes and eases the formation of biofilm increases due to the occurrence of cell-to-cell interaction [70]. This results in advantageous characteristics for the CRISPR nucleases, such as delivering molecules. These molecules change the microbial communities’ composition in biofilms [71].

A cis-conjugative mechanism was developed in plasmids which helps in encoding the CRISPR nucleases as well as conjugative machinery [72]. A total of 65 sgRNA were developed with the following non-essential genes 23, genes with unknown phenotypes 38, and significant gene 4 which were targeted. Upon growth in such a media which eases the cell-to-cell interaction, it was discovered that the plasmids which possess conjugative machinery, as well as CRISPR nuclease, pertain to a greater level of conjugative transmission from *E. coli* to *Salmonella enterica*. Compared with the critical genes, many single or multiplexed sgRNAs that target non-essential genes are involved in the destruction of *S. enterica*. The set of non-essential genes which result in the stated consequences are inclusive of askatG (catalase reductase), gltJ (glutamate/aspartate transporter), yghJ (putative lipoprotein), and aegA (putative oxidoreductase).

Additionally, the demonstration was presented that receivers of the cis-conjugative plasmids transform to be potent donors ensuing for conjugation rounds. This results in an exponential increment in the number of conjugative donor bacteria in the population/community. The combination of the distribution method and the CRISPR nucleases results in a viable option for microbiota transformation. The treatment of the entero-hemorrhagic strain of *E. coli* (EHEC) is carried out through the respective phagemid; eae targeting intimin, an encoded virulence factor of the *E. coli* O157:H7. This is required for pathogenesis in the intestine and colonization purposes. The phagemid caused about a 20-fold decline in cell counts [57]. The results help us to conclude that the work process of CRISPR-Cas will assist in the eradication of selective genomes. Theoretically, this will help in the reduction of undesirable genes such as antibiotic tolerance, virulence loci, or the pathway of metabolism in the bacterial community so the prevention of bystanders. The method can also be subjected to the utilization of alteration in the composition of various bacterial species.

Recent therapies that practice medicine to change human microbiota, prebiotics, or probiotics can potentially alleviate disease symptoms. However, these therapies are still little understood in terms of their work processes. An in-vivo mouse model was used to investigate the efficacy of the Cas9 phagemid against infections [39]. Topical therapy of the backs of CD1 mice infiltrated by RNK cells with the CRISPR-Cas9 antimicrobial pDB121: aph phagemid (against the kanamycin resistance gene aph) [73] resulted in a substantial reduction in the proportion of RNK cells, which was substantially distinct from 2% mupirocin or 200 mg per mouse streptomycin [39].

Citorik et al. found that treating EHEC with cas9-eae phagemid enhanced survival in *Galleria mellonella* larvae substantially more than no care control. This treatment was significantly more effective than the carbenicillin treatment, to which *G. mellonella* was susceptible. The EHEC treatment with cas9-eae phagemid was also more effective than chloramphenicol treatment, to which *G. mellonella* was resistant [57]. In another analysis, Kiga et al. created a Cas13a-based phage to attack carbapenem-resistant *E. coli* and methicillin-resistant *S. aureus*. These results encourage the use of the CRISPR-Cas method and phagemid as possible substitutes for bacterial strains that are extremely resilient to currently available antimicrobials [74]. However, there are two major drawbacks to using phagemid. To begin with, phagemid does not create additional phages after infection, implying that the amount of phagemid required for therapy is substantially greater than the target population’s size. Second, the restricted host variety and large-scale populace of phagemids can prevent widespread adoption. A benefit of programmable Cas9-mediated killing is the ability of a nuclease with two or more crRNA guides to activate distinct plasmid and/or chromosomal sequences, which might minimize unaffected clones that desert phagemid care by creating target mutants, as well as expand the number of targeted cells. Furthermore, delivering the sequence-specific Cas9 nuclease and reprogramming it to target other sequences decreases plasmid substance in a bacterial population devoid of destroying the cells, preventing non-pathogenic strains from antibiotic-resistant and/or virulence plasmid transition.

Many scientists concentrated on the conjugative-, phage- and polymeric nanoparticle-based CRISPR delivery methods as they are found to be the most efficient delivery methods. These delivery methods use the best fatal competent goal genes as potential pathways. In the case of planktonic and biofilm ecosystems, the best delivery method is one including a mixture of polymeric, phage, and conjugative nanoparticles. Moreover, lacking certain techniques to cope with pathogenic species and the whole removal of the target organism or fluctuating the composition of microbial communities are significant problems in microbiology, infectious disease control, and microbiome modification [75,76,77]. While CRISPR-based nucleases can potentially decrease the comparative profusion of the target and infection-causing bacteria by providing sequence-specific antimicrobial drugs, producing a widely functional and powerful delivery mechanism persists as a significant hurdle. CRISPR-Cas delivery vectors or vehicles, as well as their architecture, diverse bacterial communities, various modes of tolerance to one antibacterial agent in the organisms, the risk of mutations in target genes, legislation, and social roles of CRISPR-Cas-based antibacterial agents, could all be issues in the future.

## 6. Applications of CRISPR-Cas9 System

The CRISPR-based system has a lot of applications in curing diseases, correcting mutations, and improving crop quality and production. The detail is given below (Figure 2, Table 3).

### 6.1. Correction of Gene Mutations

Correcting recessive dystrophic epidermolysis bullosa (RDEB) through iPS (induced pluripotent stem) cells involves applying CRISPR/Cas9-based targeted. It is incredibly efficient and safe for gene correction. The hi-fi Cas9 (SpyFiCas9) nuclease has evident genome-wide off-target effects [78]. Targeted gene modification through CRISPR/Cas9 is an efficient way to evaluate gene function and accurately employ cellular behavior and process. Investigators can use GMOs (genetically modified organisms) to comprehend further the etiology of various disorders and elaborate the biochemical pathway utilized for an improved therapeutic strategy. These genome editing methods have helped to eradicate lethal diseases. CRISPR/Cas9 technology is used to produce chimeric antigen receptor T cells to damage malignant cells [79]. The CRISPR/Cas9 method has been used to successfully correct genetic disorders in mice or cystic fibrosis patients’ intestinal stem cell organoids. In an adult mouse model of human hereditary tyrosinemia disease, the Fah mutation has been corrected through CRISPR/Cas9 method. In this way, the symptoms of the disease have been eradicated [80].

### 6.2. Infectious Disease Applications

Infectious disease applications have been expanded through the CRISPR-Cas system. This technology promises to explain basic host-microbe relationships, help in the advancement of fast and precise diagnostics, and improve infectious disease prevention and care.

Knowing how bacteria, viruses, fungi, and parasites cause disease in humans is critical for providing the best health treatment and rationally designing tailored treatments and vaccinations. CRISPR Cas9-based genome editing is utilized to understand gene and protein connections to the molecular pathogenesis of a variety of pathogens.

Early detection, as well as prevention of infectious diseases, is facilitated by rapid and reliable diagnostic testing, which allows better clinical care and the prompt application of infection management and various other public health interventions to reduce disease transmission. A perfect fast diagnostic test will be responsive and specific, simple to administer and translate, compact, and inexpensive, allowing it to be used in a variety of clinical environments, including those with minimal resources. The CRISPR-Cas has aided in the advancement of fast and precise diagnostics for infectious diseases.

Many researchers are using CRISPR-Cas9 to improve diagnostics for infectious diseases. A combined nucleic acid sequence-based augmentation system known as NASBA is an example of isothermal amplification. This method is used in combination with CRISPR-Cas9 to differentiate between Zika virus strains that are closely related [80]. After applying a synthetic stimulation sequence to NASBA-amplified viral RNA, the researchers used a Cas9 and sgRNA complex to slice the resulting dsDNA. The presence or absence of a strain-specific PAM stemmed in Cas9-cleaved DNA fragments that were either abridged or full-length strands. The triggered turn was activated by full-length strands but not by truncated strands, resulting in a color shift on a paper disc and stable strain distinction [81].

### 6.3. Revolutionizing Fungal Disease Control with CRISPR-Cas9

Fungal infections are a serious global health issue because they may cause various diseases in plants, animals, and humans. Fungicides have been used to treat fungal diseases in the past, but they have the potential to damage the environment and breed resistant fungi. CRISPR-Cas9 technology is used in this case as a last resort. A gene-editing technique called CRISPR-Cas9 enables the precise modification of certain genes [82]. With the use of this technology, scientists may target particular genes in fungi and prevent them from being able to spread infection. This method is extremely specialized and can only target the genes that are accountable for the fungal pathogen’s virulence, leaving untargeted genes unaffected [83].

The use of CRISPR-Cas9 technologies to manage fungus infections has shown considerable potential. One use is the direct targeting and editing of fungal genes required for growth, pathogenicity, or drug resistance [84]. Researchers have successfully killed or inhibited the growth of harmful fungi by turning off these genes, perhaps improving treatment outcomes. Recently, the genomes of *Candida albicans* [85,86], *Aspergillus* [87], and *Cryptococcus* [88] were edited using the CRISPR/Cas9 system. This work has the potential to advance our understanding of the molecular causes of fungal infection and antifungal drug resistance.

The study of Vyas et al. (2015) employed CRISPR-Cas9 to deactivate a gene required for virulence in the fungus *Candida albicans*. The fungus’s EFG1 gene plays a role in controlling the expression of other virulence genes. In a mouse infection model, the researchers’ disruption of EFG1 greatly decreased the virulence of *C. albicans* [85].

In addition to being more precise than older approaches to fungus control, CRISPR-Cas9 technology offers additional benefits. Contrary to fungicides, which need to be administered repeatedly, CRISPR-Cas9 technology may permanently change the fungus’s DNA, preventing it from spreading illness. In the long term, this strategy may also be more cost-effective because it eliminates the need for fungicides and other conventional means of preventing the growth of mold [89].

Another potential use of CRISPR-Cas9 is in developing new antifungal agents. The approach may be used to find drugs that kill fungus by either targeting certain genes or by interfering with essential cellular functions. In addition, CRISPR-Cas9 can be utilized to create fungus strains that are less aggressive or resistant to antifungal medications, which may lower the risk of infections and enhance treatment results [84,90].

Conclusively, CRISPR-Cas9 technology offers enormous potential for preventing and treating fungal infections in agricultural contexts and scenarios involving human health. With the potential for enduring impacts, this technique provides a highly specialized and possibly economical way to control fungi. While more research is needed to fully realize the potential of this technology, the use of CRISPR-Cas9 to control fungal infections represents an exciting and promising development in the field of disease control.

### 6.4. Emerging Therapeutic Applications

Although all bacteria do not use CRISPR-Cas systems, growing evidence supports their role in blocking the gaining of the genomic elements which impart antibiotic resistance, improving the likelihood that bacteria’s defenses could be used therapeutically against them Figure 3 depicts the CRISPR approaches towards microbiome therapies. Researchers have proved that the I-F CRISPR system in *E. coli* is present in *E. coli* which is linked with antibiotic sensitivity. CRISPR technology has been suggested as a way to grow specifically titratable antimicrobials to eradicate pathogens. This concept was used in vitro to kill single strains of *E. coli* as well as *Salmonella enterica* in pure and mixed culture experiments by utilizing a subtype I-E CRISPR-Cas system. RNA-guided Cas9 system was used by Bikard et al. that was transmitted by phagemid killed virulent. However, it did not kill the avirulent strains of *S. aureus*; without destroying the host bacteria, it eliminated plasmids containing the mecA methicillin resistance gene [39].

Oral vaccines are being developed through *Saccharomyces boulardii* which is engineered with the help of the CRISPR-Cas system [91].

### 6.5. Role in Gene Expression

CRISPR/Cas9 is a useful tool in genetic engineering. It has been used in epigenetic studies to elicit gene expression. In a study, two methods were introduced to selectively regulate DNA methylation at the selective CpG site via utilizing CRISPR/Cas9 method. In this way, the gene expression was induced successfully [91]. The property of CRISPR/Cas9 to edit the gene has revolutionized cell therapy. The gene editing property of the CRISPR/Cas9 system is being improved and optimized through the use of artificial nucleic acid molecules (ANAMs) in cancerous cells. It is successfully proved that ANAMs improve transgene expression by inhibiting innate immune response (IIR) in the cells [91].

## 7. Challenges

It is proven that bacteriophages with their bacterial hosts play a significant role in maintaining a healthy gut microbiome in healthy humans [92]. However, CRISPR/cas9 system can lead to microbiota dysbiosis. CRISPR-Cas-based dealings resolve this problem by the addition of probiotics to preserve microflora homeostasis [93]. Therefore, the gRNA strategy and choice approach for the target site must be inflexible. CRISPR/cas9 system can initiate many off-target consequences, resulting in harmful results such as mutations. To solve this issue, the software is designed to minimize off-targeting and enhance on-target efficacy. Examples of this software that can be included are WTSI genome editing (WGE), E CRISPR, RGENs tools, GT-Scan, and genome engineering resources [94]. Another problem is that the majority of phage species have a limited host selection and tolerance to this “CRISPR-Cas” mechanism has evolved by deletions or mutations. The effectiveness of this antimicrobial solution in matrices for successful treatment and delivery remains a major challenge since its formulation. Another challenge is that these CRISPR-cas system methodologies deal only with the target cell and cannot cope with the limitations linked with the delivery of this system to respond to intracellular infection.

DNA repair machinery is activated through a double-strand disruption that is produced via Cas9. This molecular mechanism is utilized to insert DNA fragments (such as cDNAs). Since the DNA repair system’s job isn’t to add DNA fragments to the genome, the targeted allele also has extra modifications including deletion, incomplete or multiple integrations of the targeting vector, and duplication [95]. Standard ES cell-based ventures are often plagued by secondary unwanted mutational events at the target locus, and researchers have worked out how to prevent developing mice with passenger mutations. To recognize the correct recombination events in ES cells, most laboratories use a range of positive and negative selection procedures, as well as confirmation procedures aimed at identifying added mutations at the target site. When the CRISPR/Cas9 technique is applied directly to embryos, it is known as CRISPR/Cas9. However, selecting the desired case is problematic, significantly limiting the chances of finding the desired allele. Mosaicism was also observed in founder mice created using the CRISPR/Cas9 system. In addition, mosaicism was discovered in founder mice created using the CRISPR/Cas9 method. This method renders it incredibly difficult to identify unexpected genomic mutations at the target site [96].

## 8. Future of CRISPR

As an alternative to being utilized as a traditional broad-spectrum antibiotic, the main focus of CRISPR-cas9 is on maintaining the structure and composition of the microbial population. Future models will be capable of forecasting not just the performance but also the result of CRISPR-Cas9 editing. Scientists will be capable of removing individual DNA fragments and, in this way, monitor the consequences of CRISPR-Cas9 editing. Furthermore, a recent study discovered that the mutation caused by CRISPR-Cas9 cleavage repair was not spontaneous and was revealed through the target sequence [97]. As a result of this discovery, researchers could forecast the mutational consequence of CRISPR-Cas9 editing, enabling them to create accurate edits without using knock-ins [98].

CRISPR–Cas could be reclaimed for other uses, for example, editing mitochondrial and chloroplast genomes, mapping cell lineage to deduce the pattern underlying plant growth, designing the genetic circuits to combine and transfer signals, creating plant biosensor, and many other plants’ synthetic biology uses. Overall, the CRISPR–Cas system has transformed and will continue to transform gene therapy, disease diagnosis, and agriculture and plant biotechnology.

## 9. Conclusions

In conclusion, the CRISPR/Cas9 system has shown great potential in combating antimicrobial resistance by targeting and cutting plasmids carrying antibiotic-resistance genes or by destroying specific genes in bacteria responsible for resistance mechanisms. While pre-clinical studies have yielded promising results, challenges such as enhancing the system’s efficiency and specificity, developing effective delivery systems, and addressing safety concerns remain to be addressed. Public awareness regarding antibiotic usage is also crucial. Although ongoing research is expected to yield promising results, ethical and regulatory concerns must be addressed as well. Additionally, the use of CRISPR could potentially lead to the evolution of new antibiotic-resistant strains of bacteria.

## Figures and Tables

**Figure 1 antibiotics-12-01075-f001:**
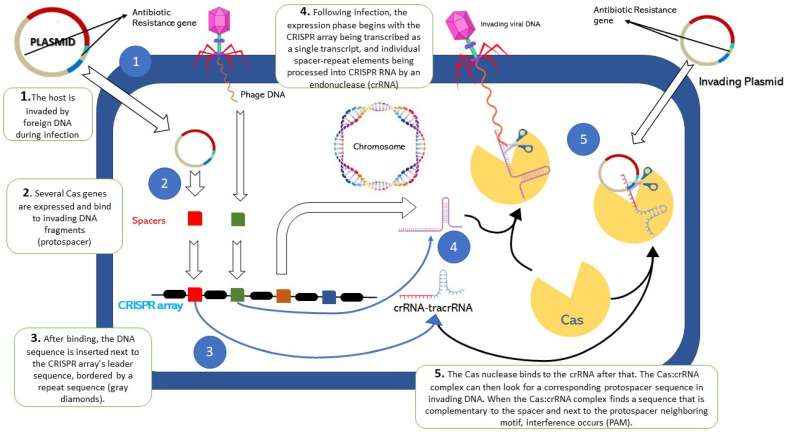
Clustered Regularly Interspaced Short Palindromic Repeats (CRISPR)-Associated (CRISPR-Cas) and bacterial antibiotic virulence. 1. The host is invaded by foreign DNA during infection. 2. Spacer acquisition. 3. Incorporation of spacers in CRISPR array. 4. Formation of crRNA and transRNA complex containing specific spacers. 5. CrRNA guide *cas* genes to target invading DNA.

**Figure 2 antibiotics-12-01075-f002:**
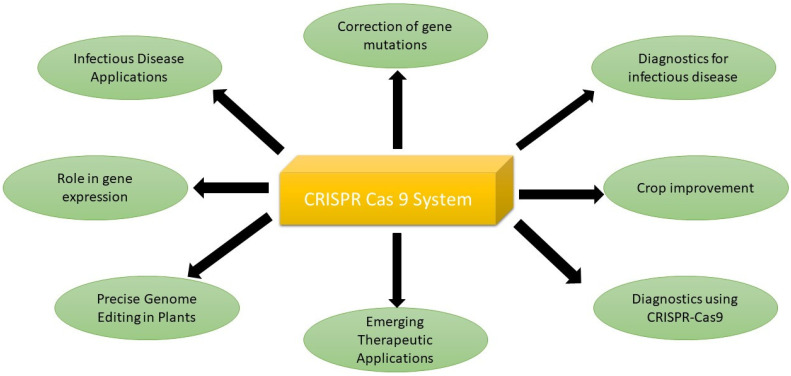
Applications of CRISPR-Cas9 system.

**Figure 3 antibiotics-12-01075-f003:**
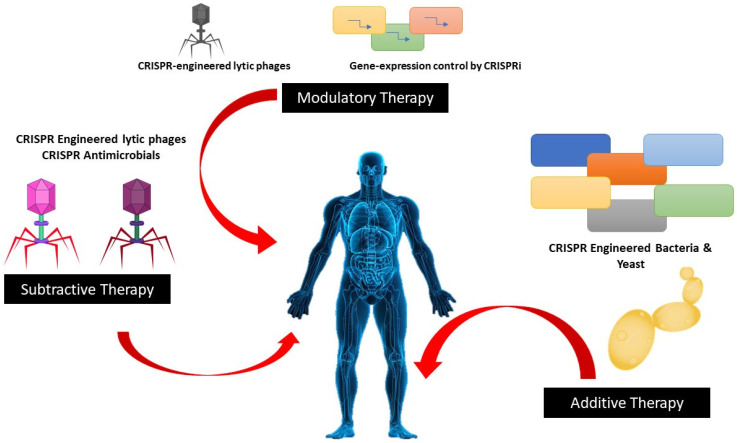
CRISPR approaches microbiome therapies.

**Table 1 antibiotics-12-01075-t001:** Classification, Standard features, and Characteristics of each CRISPR-Cas system.

Class	Type	Subtype	Signature Genes	Target	Effector	Families of Encoded Proteins	CRISPR Protein	Associated Type-Subtype	Function	References
1 (A multi-Cas protein)	I	A, B, C, D, E, F, U	I-A; Cas8a2, Cas5 I-B; Cas8b I-C; Cas8c I-D; Cas10d I-E; Cse1, Cse2 I-F: Csy1, Csy2, Csy3, Cas6f	dsDNA	Cascade	COG1518	Cas 1	I, II, III-A, III-B, IV, possibly VI	DNA Nuclease	[15]
1	III	A, B, C, D	III-A: Csm2, Csx10 all1473 III-B Cmr5, MTH326 (Cas10 or Csx11)	ssRNA	Cascade	COG1203	Cas 2	I, II, III-A, III-B, V, some VI	RNA Nuclease	[15]
1	IV	A, B	DinG (Csf4)	dsDNA	Cascade	COG1468	Cas 3	I	DNA nuclease and helicase	[15,22]
1	II	A	Csn-2	dsDNA	SpCas9	COG1343	Cas 4	Mostly type I, II & V	DNA nuclease	[15,24]
2	II	A		dsDNA	SpCas9	COG3512	Cas 5	Type I, I	Ribonuclease responsible for converting pre-crRNA to mature crRNA.	[24]
2	II	B	Cas9 (Csx12 subfamily)	dsDNA/ssRNA	FnCas9	COG1343 and COG3512	Cas 6	Most type IIIB and type I	Pre-crRNA is converted to mature crRNA by ribonuclease.	[24]
2	II	C	N/A	dsDNA	NmCas9	COG1343 and COG3512	Cas 7	I, III, IV	It binds with crRNA and comprises of an RNA recognition motif.	[5]
2	V	A	Csm4, Csx10, Cmr3	dsDNA	Cas12a (Cpf1)	COG1688 (RAMP)	Cas 8	Most type I	It forms effector complex large subunit in type I	[25]
2	V	B	Cas5, Csy2	dsDNA	Cas12b (C2c1)	COG1688 (RAMP)	Cas 9	II only	DNA nuclease	[26]
2	V	C	Csc1, Csf3	dSDNA	Cas12c (C2c3)	COG1688 (RAMP)	Cas 10	Some type I, most type III	It forms effector complex large subunit in type III.	[25]
2	VI	A	-	ssRNA	Cas13a (C2c2)	COG1583 and COG5551 (RAMP)	Cas 12 (cpf1)	V	crRNA sorting, DNA nuclease	[25,27]
2	VI	B	Cmr6	ssRNA	Cas13b (C2c4)	(RAMP)	Cas 13 (C2c2)	VI	crRNA sorting, RNA nuclease	[15,28]
2	VI	C	-	ssRNA	Cas13c (C2c7)	(RAMP)	Csm, Cmr	III	Nucleases for single-stranded DNA and RNA	[15,28]
2	VI	D	-	ssRNA	Cas13d	(RAMP)	RNase III	II	tracrRNA is processed, and crRNA maturation is aided by this system.	[22]

**Table 2 antibiotics-12-01075-t002:** Potential targeted therapeutics for CRISPR-Cas systems.

CRISPR System	Organism	Gene Target	Reference
Type I-E (Cas3)	*E. coli*	bla_NDM-1_, bla_CTX-M-15_	[37]
Type II (Cas9)	*E. coli*	Bla_TEM_, bla_SHV_	[38]
Type II (Cas9)	*S. aureus*	aph-3, mecA	[39]
Type II (Cas9)	HIV-1	LTR U3 region	[40]
Type II (Cas9)	HIV-1	LTR, gag, pol	[41]
Type II (Cas9)	HSV-1	EBNA-1, OriP	[42]
Type II (Cas9)	HBV	Repeat region of integrated genome	[43]
Type II (Cas9)	HPV	E6, E7	[44]

**Table 3 antibiotics-12-01075-t003:** Application of different CRISPR-Cas systems.

CRISPR-Cas	Mechanism/Function	Delivery Vehicle
CRISPR-SpCas9	single-RNA-mediated DNA endonuclease	Adenoviral vector Lentiviral vector Retroviral vector
CRISPRi	single-RNA-mediated inhibition of mRNA transcription	Lentiviral vector Retroviral vector
CRISPRa	single-RNA-mediated activation of mRNA transcription	Lentiviral vector
CRISPR-SaCas9	single-RNA-mediated DNA endonuclease	AAV vector Lentiviral vector
FnCas9	single-RNA-mediated PAM-independent inhibiting of translation of target RNA	pcDNA3.3 vector
C2c1/3	dual-RNA-guided DNA endonuclease	No mammalian expression vector

## Data Availability

All data generated or analyzed during this study are included in this published article.

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
