# Peer review of "CRISPR-Cas9 System: A Prospective Pathway toward Combatting Antibiotic Resistance"

_antibiotics, 2023, doi:10.3390/antibiotics12061075_

Round 1
Reviewer 1 Report (Previous Reviewer 2)
The quality of manuscript is improved.
Author Response
Dear Colleague
Thank you very much for sparing your valuable time for reviewing our MS and your feedback. We are grateful.
Reviewer 2 Report (New Reviewer)
The manuscript is an interesting review that describes ideas, methods, and techniques that could potentially enable the CRISPR-Cas9 system to prevent antibiotic resistance in bacteria. The manuscript contains also other information about CRISPR-Cas9 such as a brief description of the working mechanism and possible applications in the cure of diseases. In general, this work is well-written and worth reading.
I would only suggest the authors write in italics all names of organisms and Latin words in the text.
On line 75, "that is consists", is shall be deleted.
Author Response
The manuscript is an interesting review that describes ideas, methods, and techniques that could potentially enable the CRISPR-Cas9 system to prevent antibiotic resistance in bacteria. The manuscript contains also other information about CRISPR-Cas9 such as a brief description of the working mechanism and possible applications in the cure of diseases. In general, this work is well-written and worth reading.
I would only suggest the authors write in italics all names of organisms and Latin words in the text.
On line 75, "that is consists", is shall be deleted.
Response: Thank you for taking the time to review the manuscript and for your valuable feedback. We have taken note of your suggestion to write all names of organisms and Latin words in italics, and we will incorporate this into the final version of the manuscript.
We also appreciate your comment about the typo on line 75, and we will remove the unnecessary "is" from the sentence.
Reviewer 3 Report (New Reviewer)
The abstract has not sufficiently described the full content of the review, rather, it focuses on perspectives.
Introduction
Line 32-33: I propose …developing resistance mechanisms against …. Instead of “genes” as theses can be acquiring new genes, missing existing genes, single mutations and many other recombination mechanisms.
Line 34: what are the authors using single quote?
Line 39: discovery sound better than arrival
Lion 40: .. Have evolved over time: do you mean parallelly? Please highlight it, as it is important.
Line 44: even though I understand what the authors meant, this can mislead. First, just put ARGs into (), you only use “also known as” when it is an alternative and another definition of the concept. Second, ARG, by itself is not a major public health issue, it is natural as you later state. Please rephrase it.
Line 67: are there any other virus attacking bacteria than phages?
In the overview I recommend to the authors to provide a short background about the CRISPR system, when and how it was discovered, how it is acquired generally and now talk about the CRSPR-cas operons, and the cas9 specifically. Please reorganize the three paragraphs.
Line 89-90: some antibiotic resistance has always existed (especially genes) and most of the time horizontally acquired from neighbour bacteria communities, others are results of recombination (evolution) like mutations etc… The concept “emerge” should be clarified
Line 96: not development, it is an acquisition.
Line 101: I am not sure about this statement. “Ignored”?
Line 103: yes, there is an association between ESBL and cephalosporin resistance, but ESBL doesn’t confer resistance to cephalosporin.
Line 104: Bacteria name should be in italic
Line 115-116: is there an association between lack of medical facilities and antimicrobial resistance?
Line 127-129: please provide reference
Line 154: “donnot” not “donot”
I don't see the importance of this section 5
Line 201-203: not antibiotic resistance to start with. It regulates acquisition of external genetic material that includes genes coding for antimicrobial resistance. Please rephrase the statement.
Section 6 and 7 should be merged
I haven’t seen table1 but in table 2, I don’t understand why you have to include virus a target organism while you are dealing with antimicrobial resistance.
Line 244: The presence of E. coli was shown by Touchon et al [45]; not clear to me
Line 246: write number, not “no”
Line 246-249: Tn3 are not spacers, or the statement is not understandable.
Overall, the real mechanism that underline the role of CRISPR system in the antimicrobial resistance is not clearly highlighted.
Can’t see figure 1
Figure 2: phase 1: …during infection…. Please provide footnote with clear description
Also, CRISPR array (repeats and spacers) are inserted in the bacteria chromosome.
Line 405: mechanism or “work process”?
Line 423: why do you have to create a new paragraph while you are still describing the same thing?
Line 432-445: the authors stated with “Many scientists” but with only one citation in the whole paragraph
In the section: Application of CRSIPR-cas9 system, I think the authors should focus on antimicrobial resistance mechanism and therapy. Too much information drown the information.
Your conclusion should also focus on summarizing the mechanism and the perspective, not the current trend.
Author Response
The abstract has not sufficiently described the full content of the review, rather, it focuses on perspectives.
Response: We apologize for not adequately describing the full content of the review. we have revised the abstract to include a brief summary of the key findings and main points covered in the review, in addition to the perspectives presented.
Introduction
Line 32-33: I propose …developing resistance mechanisms against …. Instead of “genes” as theses can be acquiring new genes, missing existing genes, single mutations and many other recombination mechanisms.
Response: We completely agree with your suggestion and modified the language from "genes" to "resistance mechanisms" when discussing the development of resistance in organisms.
Line 34: what are the authors using single quote?
Response: We used single quotes to indicate the term 'antibiotic resistance' being discussed in line 34 as the term is less than four lines long.
Line 39: discovery sound better than arrival
Response: We have changed the word 'arrival' to 'discovery', which is a more appropriate word choice in this context.
Line 40: Have evolved over time: do you mean parallelly? Please highlight it, as it is important.
Response: To clarify, when I said that resistant bacteria have evolved over time, I meant that they have evolved independently of the development of antimicrobials. While the discovery of antimicrobials in the early nineteenth century played a significant role in the fight against bacterial infections, the evolution of bacteria and their ability to become resistant has occurred parallel to the development of these antimicrobials. Therefore, it is important to understand that the emergence of antimicrobial resistance is not solely a result of the use of antimicrobials but is also influenced by other factors such as genetics, environmental factors, and the widespread use of antimicrobials in both human and animal populations.
Line 44: even though I understand what the authors meant, this can mislead. First, just put ARGs into (), you only use “also known as” when it is an alternative and another definition of the concept. Second, ARG, by itself is not a major public health issue, it is natural as you later state. Please rephrase it.
Response: We have made the following revisions to address your concerns:
Line 44: We have removed the phrase "also known as" as it is unnecessary in this context. We have also clarified that ARGs are a natural occurrence and not a major public health issue. Our revised sentence now reads as follows: “Antimicrobial resistance genes (ARGs), which are naturally occurring genetic elements, play a crucial role in the development and spread of antibiotic resistance.”
Line 67: are there any other virus attacking bacteria than phages?
Response: Yes, apart from phages, other viruses can also infect bacteria. Virophages are one such example, which replicates within another virus that is already infecting a bacterial host. Virophages are believed to have a regulatory role in controlling viral populations in aquatic environments. Virus-like particles (VLPs) are non-infectious particles that resemble viruses but cannot replicate or infect cells. Bacteriocins, on the other hand, are peptides or proteins produced by certain bacteria that can kill other bacteria. While bacteriocins are not technically viruses, they function as an antibacterial weapon that some bacteria use to compete with others for resources.
In the overview I recommend to the authors to provide a short background about the CRISPR system, when and how it was discovered, how it is acquired generally and now talk about the CRSPR-cas operons, and the cas9 specifically. Please reorganize the three paragraphs.
Response: We have provided a short background about the CRISPR system, when and how it was discovered, and how it is acquired generally, and now talk about the CRSPR-cas operons and the cas9 specifically in three paragraphs.
Line 89-90: some antibiotic resistance has always existed (especially genes) and most of the time horizontally acquired from neighbour bacteria communities, others are results of recombination (evolution) like mutations etc. The concept “emerge” should be clarified.
Response: In this case, we were referring to the appearance or increase in prevalence of antibiotic-resistant strains of bacteria in a particular population or environment. As you correctly noted, many antibiotic resistance genes have existed in bacterial populations for a long time, but their emergence or increased prevalence can occur through mechanisms such as horizontal gene transfer, genetic recombination, or selection pressures from antibiotic use.
Line 96: not development, it is an acquisition.
Response: We have changed the word development to “acquisition”.
Line 101: I am not sure about this statement. “Ignored”?
Response: We agree that the use of the word "ignored" may not accurately reflect the reality of the situation. Therefore, we have modified it and the sentence now reads as follows: “Antibiotic resistance was not initially recognized as a serious problem, despite the potential consequences it could have on public health.”
Line 103: yes, there is an association between ESBL and cephalosporin resistance, but ESBL doesn’t confer resistance to cephalosporin.
You are correct that the presence of ESBL in bacteria is associated with increased resistance to cephalosporins, but it does not directly confer resistance. ESBLs are enzymes that are capable of breaking down certain types of antibiotics, including cephalosporins, making them ineffective. Therefore, the presence of ESBLs in bacteria can lead to cephalosporin resistance, but it is not the direct cause of resistance. Thank you for pointing this out, and I hope this clarifies any confusion.
Line 104: Bacteria name should be in italic
Response: Changes made accordingly in line 104.
Line 115-116: is there an association between lack of medical facilities and antimicrobial resistance?
Response: While the relationship between lack of medical facilities and antimicrobial resistance is complex and multifactorial, there is evidence to suggest that inadequate access to healthcare and poor sanitation and hygiene facilities can contribute to the development and spread of antimicrobial resistance.
Line 127-129: please provide reference
Response: Reference has been provided.
Line 154: “donnot” not “donot”
Response: Correction has been made.
I don't see the importance of this section 5
Response: Section 5 has been removed from the manuscript as per the suggestion of the reviewers.
Line 201-203: not antibiotic resistance to start with. It regulates acquisition of external genetic material that includes genes coding for antimicrobial resistance. Please rephrase the statement.
Response: This section has been removed from the manuscript.
Section 6 and 7 should be merged
Response: We have merged section 6 and 7.
I haven’t seen table1 but in table 2, I don’t understand why you have to include virus a target organism while you are dealing with antimicrobial resistance.
Response: Table 1 is at the end of the manuscript. The inclusion of viruses as target organisms is crucial due to the fact that antimicrobial resistance (AMR) is not restricted to bacteria alone, but also extends to viruses and other microorganisms. Viral infections are a significant global cause of mortality and morbidity, and the emergence of antiviral resistance is a growing concern. Additionally, the overuse and misuse of antimicrobial agents, such as antibiotics and antivirals, often used interchangeably in clinical practice, contribute to the development and spread of AMR. Consequently, it is necessary to account for all types of microorganisms, including viruses, when dealing with AMR.
Line 244: The presence of E. coli was shown by Touchon et al [45]; not clear to me
Response: We rephrased the sentences to make it clear and it reads as follows:
Touchon et al [46] found that the existence of plasmids, integrons, or antibiotic resistance has little effect on number of repeats in E. coli CRISPR loci.
Line 246: write number, not “no”
Response: We have made correction accordingly.
Line 246-249: Tn3 are not spacers, or the statement is not understandable.
Response: To make the statement understandable, we rephrased it and the text now reads as follows: “They also found no correlation between the presence of spacers matching antibiotic resistance genes or elements associated with the movement of antibiotic resistance genes, such as Tn3, intI, ISEcp1, and ESBL production, ESBL type, or replicon. Additionally, no spacers matching plasmid sequences were detected.”
Overall, the real mechanism that underline the role of CRISPR system in the antimicrobial resistance is not clearly highlighted.
Response: In response to your comment, we have made revisions to the manuscript to more clearly highlight the mechanisms underlying the role of CRISPR in antimicrobial resistance. We hope that these revisions better address your concerns. Figure 2 has been modified which as well now highlights the mechanism in a better way.
Can’t see figure 1
Response: Figure 1 is within the manuscript with the description, “Statistics from the National Healthcare Safety Network demonstrates a comparison of azithromycin in 2019 and 2020.”
Figure 2: phase 1: …during infection…. Please provide footnote with clear description
Also, CRISPR array (repeats and spacers) are inserted in the bacteria chromosome.
Response:Thank you for pointing out the mistake. The necessary changes have been made in Figure 2.
Line 405: mechanism or “work process”?
Response: Made the correction by replacing “mechanism” to “work process.”
Line 423: why do you have to create a new paragraph while you are still describing the same thing?
Response: We have put all this in the same paragraph.
Line 432-445: the authors stated with “Many scientists” but with only one citation in the whole paragraph
Response: More references have been added.
In the section: Application of CRSIPR-cas9 system, I think the authors should focus on antimicrobial resistance mechanism and therapy. Too much information drown the information.
Response: Thank you for taking the time to review our paper and for your valuable feedback. We appreciate your suggestion regarding the focus of our paper on the potential of the CRISPR-Cas9 system in antibiotic resistance. We agree that this is an important topic to address, and we have made revisions to the manuscript to better emphasize this aspect of our research.
In response to your comment on the inclusion of antimicrobial resistance in our paper, we have carefully reviewed the content and have made the necessary changes to remove any irrelevant information. We hope that the revisions we have made will better justify the points we have presented and strengthen the overall message of our paper.
Your conclusion should also focus on summarizing the mechanism and the perspective, not the current trend.
Response: Conclusion has been revised.
Reviewer 4 Report (New Reviewer)
Javed et al summarize the importance of the CRISPR-CAS9 system in the control of an increased number of cases of antibiotic resistance in humans and animals. Overall, they have addressed the major issues concerning antibiotic resistance across the globe and how it has emerged as a major health concern in recent times. The review does provide a brief account of many reasons behind antibiotic resistance and how the CRISPR CAS9 system is being used as a tool to combat this issue. However, the review has many shortcomings when it comes to presentation, citations, and writing. Thera are many sections with no new information and with no citations.
Major comments
1. Authors need to proofread their manuscripts before the final submission. There are many grammatical mistakes i.e., space, italics and a few segments are in red font.
2. Genus and species names should be in italics throughout the manuscript.
3. There is a mention of Table 1 in the text, but I didn’t see it in the manuscript.
4. A short figure or table depicting different types of CRISPR CAS9 systems would be helpful for the readers.
5. It would be great if they could give more details about the other techniques used in the control of antibiotic resistance and what are their shortcomings when compared to the CRISPR CAS9 in the introduction section.
6. The heading of section 3 should be changed to something more impactful.
7. Line 89-90 authors need to add the reference.
8. LINE 114 -115, 117 change the wording doctor’s or veterinarian’s prescription and chemists’ shop to more general/scientific terms.
9. In Section 4 what do the other anti-infectives mean? The whole section is a little vague and very general in terms of information. There is no new information provided in this section with only two references.
10. Section 5 is again a lot of general information with very few citations.
11. Figure 1 - please provide the details of the source National Healthcare Safety Network.
12. Section 6 is not informative and repetitive.
13. It would be great if they could provide an account on the use of CRISPR CAS9 in controlling fungal infections also.
Author Response
. Authors need to proofread their manuscripts before the final submission. There are many grammatical mistakes i.e., space, italics and a few segments are in red font.
Response: We are grateful for your review of our manuscript and appreciate your feedback. Following your comments, we have carefully proofread the manuscript and made the necessary modifications to ensure that it meets the journal's criteria and standards. Thank you for your valuable input.
- Genus and species names should be in italics throughout the manuscript.
Response: The said changes has been made to the manuscript.
- There is a mention of Table 1 in the text, but I didn’t see it in the manuscript.
Response: Table 1 is located at the end of the manuscript following the "References" section.
- A short figure or table depicting different types of CRISPR CAS9 systems would be helpful for the readers.
Response: We appreciate your suggestion. The manuscript already includes Table 1, which provides a detailed classification of each CRISPR-Cas system, along with their standard features and characteristics. This table can be found at the end of the manuscript, following the "References" section.
- It would be great if they could give more details about the other techniques used in the control of antibiotic resistance and what are their shortcomings when compared to the CRISPR CAS9 in the introduction section.
Response: Thank you for the suggestion. We have added the following paragraph regarding this:
‘Other techniques like PNA antibiotics, ZFNs, and phage therapy show potential in combating antibiotic resistance, though they still have some significant limitations. PNA antibiotics, for instance, have restricted target specificity and can potentially damage healthy microbiota [7]. Phage therapy, on the other hand, necessitates the discovery and use of certain bacteriophages to target resistant bacteria, which makes it a labor-intensive and complicated operation [8]. ZFNs also have restricted target specificity and can cause off-target effects, leading to unintended genetic changes [9]. On contrary, CRISPR-Cas9 has proven to be a potential technique in case of antibiotic resistance because of its efficient genome editing capabilities and highly specific targeting. It permits precise bacterial gene modifications, including those responsible for antibiotic resistance. CRISPR-Cas9 is also very adaptable and can be used to target a variety of bacterial strains, making it a highly versatile tool in combating antibiotic resistance [10].’
- The heading of section 3 should be changed to something more impactful.
Response: We appreciate your suggestion. The heading of the section has been changed to ‘ The Perilous Persistence of Drug resistance.’
- Line 89-90 authors need to add the reference.
Response: Thank you for the comment. The reference has been added as per the suggestion.
‘Levy, S.B.; Marshall, B. Antibacterial Resistance Worldwide: Causes, Challenges and Responses. Nat. Med. 2004, 10, S122–S129.’
- LINE 114 -115, 117 change the wording doctor’s or veterinarian’s prescription and chemists’ shop to more general/scientific terms.
Response: Thank you for the suggestion. We have revised the sentences as follows:
‘Antibiotic resistance may be on the rise as a result of failing public health services, the proliferation of infectious infections, and the misuse of antibiotics without proper medical guidance [19].’
‘Furthermore, unrestricted access to antibiotics in pharmacies, coupled with a lack of consumer knowledge on the consequences of misuse, has led to a prevalence of self-medication practices [20].’
- In Section 4 what do the other anti-infectives mean? The whole section is a little vague and very general in terms of information. There is no new information provided in this section with only two references.
Response: Section 4 has been removed from the manuscript.
- Section 5 is again a lot of general information with very few citations.
Response: thank you for your suggestion. We have removed section 5 from the manuscript.
- Figure 1 - please provide the details of the source National Healthcare Safety Network.
Response: figure 1 is not included in the manuscript as it was part of the section 5 which has been removed as well.
- Section 6 is not informative and repetitive.
Response: In response to your suggestion, we have decided to remove section 6 from the paper entirely. We believe that this revision improves the flow and clarity of our manuscript, and we hope that this change meets your expectations.
- It would be great if they could provide an account on the use of CRISPR CAS9 in controlling fungal infections also.
Response: The section has been added as per the suggestion of the reviewer.
Round 2
Reviewer 4 Report (New Reviewer)
The authors have addressed all the issues and the manuscript looks more impactful now. I would suggest one more round of proofreading since there are still some grammatical mistakes.
This manuscript is a resubmission of an earlier submission. The following is a list of the peer review reports and author responses from that submission.
Round 1
Reviewer 1 Report
In the current manuscript, the authors have reviewed the potential of CRISPR Cas9 system in combating antibiotic resistance.
The article is well structured into sections and subsections. It is within the scope of the journal. However, there are some major concerns that need to be addressed to improve the article. There are many sentences throughout the manuscript that are confusing and lack clarity. The detailed comments are below:
1) The rationale of conducting the study is not clear. There are many articles already available in literature. For instance the followings are few of them:
Aslam, B., Rasool, M., Idris, A. et al. CRISPR-Cas system: a potential alternative tool to cope antibiotic resistance. Antimicrob Resist Infect Control 9, 131 (2020). https://doi.org/10.1186/s13756-020-00795-6
Tao S, Chen H, Li N, Liang W. The Application of the CRISPR-Cas System in Antibiotic Resistance. Infect Drug Resist. 2022;15:4155-4168. Published 2022 Aug 2. doi:10.2147/IDR.S370869
Gholizadeh P, Köse Åž, Dao S, et al. How CRISPR-Cas System Could Be Used to Combat Antimicrobial Resistance. Infect Drug Resist. 2020;13:1111-1121. Published 2020 Apr 20. doi:10.2147/IDR.S247271
Duan C, Cao H, Zhang LH, Xu Z. Harnessing the CRISPR-Cas Systems to Combat Antimicrobial Resistance. Front Microbiol. 2021;12:716064. Published 2021 Aug 20. doi:10.3389/fmicb.2021.716064
Getahun YA, Ali DA, Taye BW, Alemayehu YA. Multidrug-Resistant Microbial Therapy Using Antimicrobial Peptides and the CRISPR/Cas9 System. Vet Med (Auckl). 2022;13:173-190. Published 2022 Aug 11. doi:10.2147/VMRR.S366533
Araya DP, Palmer KL, Duerkop BA (2021) Correction: CRISPR-based antimicrobials to obstruct antibiotic-resistant and pathogenic bacteria. PLOS Pathogens 17(12): e1010153. https://doi.org/10.1371/journal.ppat.1010153
There has been resistance to CRISPR-Cas antimicrobials as well.
Uribe, R.V., Rathmer, C., Jahn, L.J. et al. Bacterial resistance to CRISPR-Cas antimicrobials. Sci Rep 11, 17267 (2021). https://doi.org/10.1038/s41598-021-96735-4
2) The novelty added by this manuscript to the existing knowledge is not clear.
3) Page 1, line 36-37: The sentence is not clear. The statement holds true only in the case of pathogenic bacteria.
4) Page 1, line 40-41: Not all clinically relevant strains are resistant. The sentence needs rephrasing to improve the clarity.
5) Page 2, line 57-58: It is not clear what authors want to convey here.
6) Page 2, line 60-62: These are techniques not mechanisms.
7) Page 2, line 82: Class 2 system includes (Type II, V, and VI).
8) Page 3, line 152: The sentence requires rephrasing.
9) Page 3, line 155: “…overuse of antibiotics…” not “over resistance to antibiotics…”,
10) Page 5-7: Figures 2, 3, and 4 convey redundant information. One figure would be enough to provide the required details.
11) Page 11-14: In context of the current manuscript is it relevant to describe genome editing in plants, application in crop improvement?
12) Page 12: Subsections 9.2, 9.3, and 9.4 could be merged and rewritten to provide concise information.
13) Page 15: Conclusion section needs to be rewritten to provide a new outlook or perspective. All this is already well known.
14) Reference section: Some references have missing information like doi or page number or inconsistent format which need to be corrected. Check reference 4, 6, 10, 11, 12, 14, 16, 17, 18, 20, 21, 22, 2, 25, 26, 28, 30, 31, 33, 35, 36, 41, 43, 45, 50, 55, 57, 59, 60, 66, 67, 70, 71, 73, 74, 75, 76,77, 79, 80,81, 82, 84, 87, 90, 91, and 96.
15) Reference 92 and 93 are duplicate.
Reviewer 2 Report
CRISPR-Cas systems provide versatile tools for programmable genome editing. Thie review focuses on the applications of CRISPR-Cas in antibiotic resistance. There are some current research advances on CRISPR and antibiotic resistance, but some recent or relevant important references are not cited. The main problem with this article is that the logic is not clear, especially sections 3, 4, and 5, which use more words to describe the content that is not very relevant to the topic of the article. The other thing is that the all figures and tables are not of high quality, and do not provide effective information and are not attractive.
Please revise the title and the body of the text related to "CRISPR Cas system ...... " description, we suggest to change it to "CRISPR-Cas system ......" .